# Rotor Fault Diagnosis Method Using CNN-Based Transfer Learning with 2D Sound Spectrogram Analysis

**Haiyoung Jung** [1] [iD] **, Sugi Choi** [1] [iD] **and Bohee Lee** [2],* [iD]

1 Department of Fire and Disaster Prevention, Semyung University, 65 Semyung-ro, Jecheon-si 27136, Chungcheongbuk-do, Republic of Korea
2 Department of Electrical Engineering, Semyung University, 65 Semyung-ro, Jecheon-si 27136, Chungcheongbuk-do, Republic of Korea
* Correspondence: bhlee@semyung.ac.kr

**Abstract:** This study discusses a failure detection algorithm that uses frequency analysis and artificial intelligence to determine whether a rotor used in an industrial setting has failed. A rotor is a standard component widely used in industrial sites, and continuous friction and corrosion frequently result in motor and bearing failures. As workers inspecting failure directly are at risk of serious accidents, an automated environment that can operate unmanned and a system for accurate failure determination are required. This study proposes an algorithm to detect faults by introducing convolutional neural networks (CNNs) after converting the fault sound from the rotor into a spectrogram through STFT analysis and visually processing it. A binary classifier for distinguishing between normal and failure states was added to the output part of the neural network structure used, which was based on the transfer learning methodology. We mounted the proposed structure on a designed embedded system to conduct performance discrimination experiments and analyze various outcome indicators using real-world fault data from various situations. The analysis revealed that failure could be detected in response to various normal and fault sounds of the field system and that both training and validation accuracy were greater than 99%. We further intend to investigate artificial intelligence algorithms that train and learn by classifying fault types into early, middle, and late stages to identify more specific faults.

**Keywords:** rotor fault detection; convolutional neural networks; spectrogram; transfer learning

## 1. Introduction

Industrial automated mechanical systems are widely used in various industries as we enter the fourth industrial revolution. For example, oil piping installations world-wide have been rapidly expanding to satisfy the increasing energy needs. The pipelines have the potential for structural failure due to erosion, crack propagation, human factors, environmental factors, and other causes over time [1].

In addition, a conveyor belt device, a piece of industrial machinery, is mainly used in factories to efficiently transport parts and materials such as coal and minerals [2]. The conveyor belt system makes extensive use of rollers, which are frequently the cause of system failure because foreign objects can easily get caught between them and cause roller breakage [3]. Conveyor belt wear faults are typically identified by a way worker directly inspecting a belt at a work site. However, manually inspecting the work site is time-consuming and increases the risk of accidents such as getting caught in the conveyor belt [4].

Artificial intelligence is quickly becoming a viable replacement for the traditional methods of machine failure diagnosis [5].

Recently, research on artificial intelligence-based abnormal signal detection has become a hot issue. Spandonidis. et al. developed LSTM-A and 2D-CNN algorithms based on sound data and conducted research on leak detection in oil and gas pipelines [1].

Theodoropoulos et al. conducted a study on detecting early signs of defects in ship operating conditions through a 2D-CNN model [6]. Hou et al. developed an AE-VBGMM that combines an autoencoder and a Bayesian Gaussian model to detect malicious nodes in an IoT environment [7]. In addition, Ahn et al. conducted a study on detecting abnormal motion inside a machine through supervised learning and unsupervised learning [8]. As a result, it achieved up to 99% accuracy by overcoming the disadvantages of existing machine learning algorithm structures such as SVM and BP-NN [9]. However, the LSTM model has disadvantages in that it is difficult to extract nonlinear characteristics of data and is slow [10]. Studies on these various artificial intelligence algorithms have resulted in high performance. In particular, 2D-CNN has become a hot research topic in the field of machine defect detection based on sound data, and various Fourier transform techniques such as STFT, WT, and HHT are used in the pre-processing of sound data.

Research results on the application of these 2D-CNN can be easily found in the fields of EEG signals and ECG signals [11–13]. For example, Nahal Shahini et al. proposed that the raw EEG input is applied directly to a CNN without feature extraction or selection. This methodology could be employed in brain–computer interface (BCI) applications due to high accuracy results [14]. Amin Ullah et al. studied ECG signals-based classification of arrhythmia. They applied the 2D-CNN model for the classification of ECG signals into eight classes. The l-D ECG time series signals are transformed into 2D spectrogram images through STFT. They achieved a high accuracy of 99.11% on the classification of arrhythmia [15].

In particular, the CNN based on time–frequency signal data is used for fault diagnosis of rotating bodies used in various industrial fields. Guoqiang Li et al. proposed a two-step fault diagnosis method. The first step applied the WPT (Wavelet Packet Transform) to obtain 1D time–frequency coefficients from vibration signals, which are converted into 2D gray images. In the next step, the CNN model is applied. As a result, it was confirmed that the CNN model has superior fault diagnosis capabilities than existing machine learning-based decision trees, k-nearest neighbors, and support vector machines [16]. Meanwhile, Dip Kumar Saha et al. proposed a machine-learning approach for fault diagnosis of rotary machine element bearing. The time waveform of vibration data in the system was converted to a spectrogram using the fast Fourier transform (FFT) method. In the next step, a support vector machine (SVM), a machine learning algorithm, was applied. As a result, the developed SVM model was superior to traditional ML techniques such as KNN (k-nearest neighbor), DT (decision tree), and LDA (linear discriminant analysis), but the accuracy was only 93.9% [17]. Additionally, David et al. converted sound data into image data using time–frequency signal processing techniques such as short-time Fourier transform (STFT), wavelet transform (WT), and Hilbert–Huang transform (HHT) to diagnose bearing defects. After learning a convolutional neural network (CNN) model, it was confirmed that the failure diagnosis performance of the machine was significantly improved compared with previous studies [18]. LIANG, Pengfei, et al. converted the one-dimensional vibration signal of the bearing into a two-dimensional frequency spectrogram using the FFT technique and trained the CNN model to demonstrate that the CNN model outperformed BP and SVM models [19].

As can be seen through the literature review of various studies, CNN models are superior to prior artificial intelligence models such as machine learning-based classifiers and deep learning classification models in signal classification.

However, since these CNN-based deep learning algorithms operate on computers installed in offices with abundant computer infrastructure, it is difficult to process information generated in real time in a wide range of sites with one computer resource [20]. Therefore, it is necessary to apply a compact and efficient CNN model that can be applied in the field and operate in an embedded environment placed close to rotating equipment.

In this study, we collected the sound data of conveyor belt rollers directly at the work site and collected the sound data into a spectrogram, an image form applied with an STFT, a time–frequency analysis technique. During the next step, a CNN-based transfer learning

model was applied to the converted spectrogram to perform processes such as learning, evaluation, and machine fault prediction. Finally, the developed rotor fault diagnosis model using CNN-based transfer learning was embedded in an embedded system based on Raspberry Pi (RPi) 4 to verify the performance of the failure diagnosis system in real time. In addition, the sound detection result of each sound detection module obtained through this process is transmitted to the upper controller, and instead of the entire existing sound data, only the determination result is transmitted, thereby reducing the load concentrated on the PLC and PCs.

This study is structured as follows. Section 2 presents the theoretical background. Section 3 presents the data collection and experimental processes. Sections 4 and 5 present the experimental results and analysis, and the conclusion and future research directions, respectively.

## 2. Theoretical Background

### 2.1. Short-Time Fourier Transform

STFT is a time–frequency domain analysis method that converts a one-dimensional signal into a two-dimensional matrix suitable for the CNN model. The processing technique of STFT is summarized as follows. A window signal is extracted from the desired signal by adding a short-time window, and then the Fourier spectrogram of the window signal is calculated. A spectrogram image with a time–frequency expression of the signal can be obtained by sliding the window along the time axis [21–23]. *STFT* is expressed using Equation (1) [24–27], as follows:

$$STFT(t,\ \omega) = \int_{-\infty}^{\infty} x(\tau)\omega(\tau - t)exp^{-j\omega\tau}d\tau \tag{1}$$

where $x(\tau)$ denotes a signal function and $\omega(\tau)$ denotes a window function.

Because the actual conveyor belt sound signal is a discontinuous function, the conversion of Equation (2) is performed for the $n$th discontinuous signal $x(n)$, time $m$, frequency $l$, and the length $L$ of window function $\omega(t)$ [28].

$$STFT(m,\ l) = \sum_{n=-\infty}^{\infty} x(n)\omega(n - m)exp^{-2\pi jnl/L} \tag{2}$$

The sound signal is converted from the time domain to the time–frequency domain through this process.

In this study, we converted *STFT* using the librosa.stft method in Python's Librosa package to implement *STFT* [29].

### 2.2. Convolutional Neural Networks

Figure 1 shows the representative structure of the CNN. For input image data ($128 \times 128 \times 3$). the convolutional layer creates a feature map that extracts the features of the image, whereas the pooling layer reduces the number of operations by reducing the extracted feature map. Identified features that have undergone multiple convolution and pooling processes are finally classified in the fully connected layer [30–33].

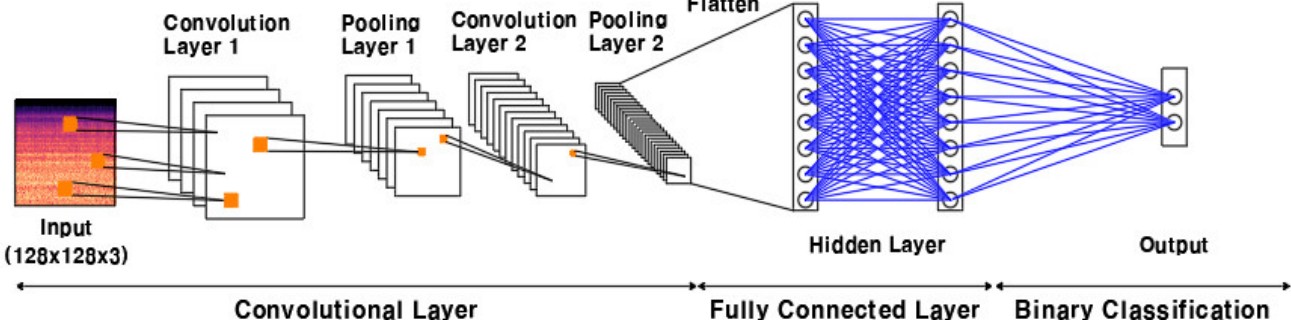

**Figure 1.** Structure of convolutional neural networks (CNNs).

In the convolution layer, the feature map from the previous layer is input into the next convolution layer. The input convolution layer applies an activation function to create the next convolution kernel.

The operation process of the convolution layer is shown in Equation (3) [31].

$$M_n{}^I = f\left(\sum X_m{}^{l-1} \times W_{mn}{}^l + B_n{}^I\right) \tag{3}$$

where *M* denotes the layer output matrix, *f* denotes the activation function, *X* denotes the previous layer input matrix, *W* denotes the kernel weight matrix, and *B* denotes the bias vector.

A crucial component of CNN is a nonlinear activation function, and the *ReLu* function is frequently used for such. The ReLu function returns zero if the input value is less than zero, and outputs the input value if it is greater than zero. Because these *ReLu* functions converge quickly and show non-saturated linearity, the learning speed is quick and the loss of slope, which is a disadvantage such as the conventional Sigmoid activation function, can be prevented. Equation (4) represents the expression of the *ReLu* function [32].

$$f(x) = ReLu(x) = max(0,\ x) \tag{4}$$

The pooling method includes max pooling and average pooling. *Max* pooling, which selects the maximum value in each area, is mainly used in the image recognition field. This pooling process has the advantage of quick calculation speed and reduces overfitting because the number of parameters to be calculated is few. In this study, we used the *Max* pooling method. *Max* pooling is calculated using Equation (5) [34], as follows:

$$P_m = MaxM_n{}^I,\ \ M_n \epsilon S \tag{5}$$

where $P_m$ denotes the output matrix and *S* denotes the size of the pooling layer.

CNNs repeatedly perform the convolutional and pooling layers before the classification step using the fully connected layer, which is the final step. The fully connected layer flattens the extracted two-dimensional array into a one-dimensional array and classifies the image using the Softmax function.

### 2.3. Transfer Learning Model

We could apply the previously learned models to new models [35] using the transfer learning model. Deep learning generally requires considerable data. In this case, the use of transfer learning can address the problems of insufficient data, time, and cost associated with data collection and labeling [36]. Table 1 shows CNN-based transfer learning models with relatively high-performance indicators of learning results compared with other transfer learning models [37].

**Table 1.** Comparison of transfer learning models.

| Model | Size (MB) | Top-1 Accuracy | Top-5 Accuracy | Parameters | Depth |
|---|---|---|---|---|---|
| Xception | 88 | 79.0% | 94.5% | 22.9 M | 81 |
| VGG16 | 528 | 71.3% | 90.1% | 138.4 M | 16 |
| VGG16 | 549 | 71.3% | 90.0% | 143.7 M | 19 |
| ResNet50 | 98 | 74.9% | 92.1% | 25.6 M | 107 |
| ResNet50V2 | 98 | 76.0% | 93.0% | 25.6 M | 103 |

In this study, we selected the Xception model because it has a deep depth, occupies a small amount of memory capacity, and has the highest validation accuracy among the transfer learning models.

To apply a transfer learning model to a new dataset, a fine-tuning technique is required to retrain a previously trained model to serve the intended purpose or to retrain some of the learned weights. Because of the fine-tuning technique, only one layer is actively learning while the others are frozen.

Figure 2 shows fine-tuning options according to various data sizes and similarities. Figure 2a shows four conditions according to the sizes and similarities of the dataset, and Figure 2b shows the transition learning model and the classifier's fine-tuning options. In this study, the entire relearning fine-tuning technique, which is the first quadrant condition with the highest validation accuracy, was selected and applied to model training after all the fine-tuning techniques of the four conditions were performed.

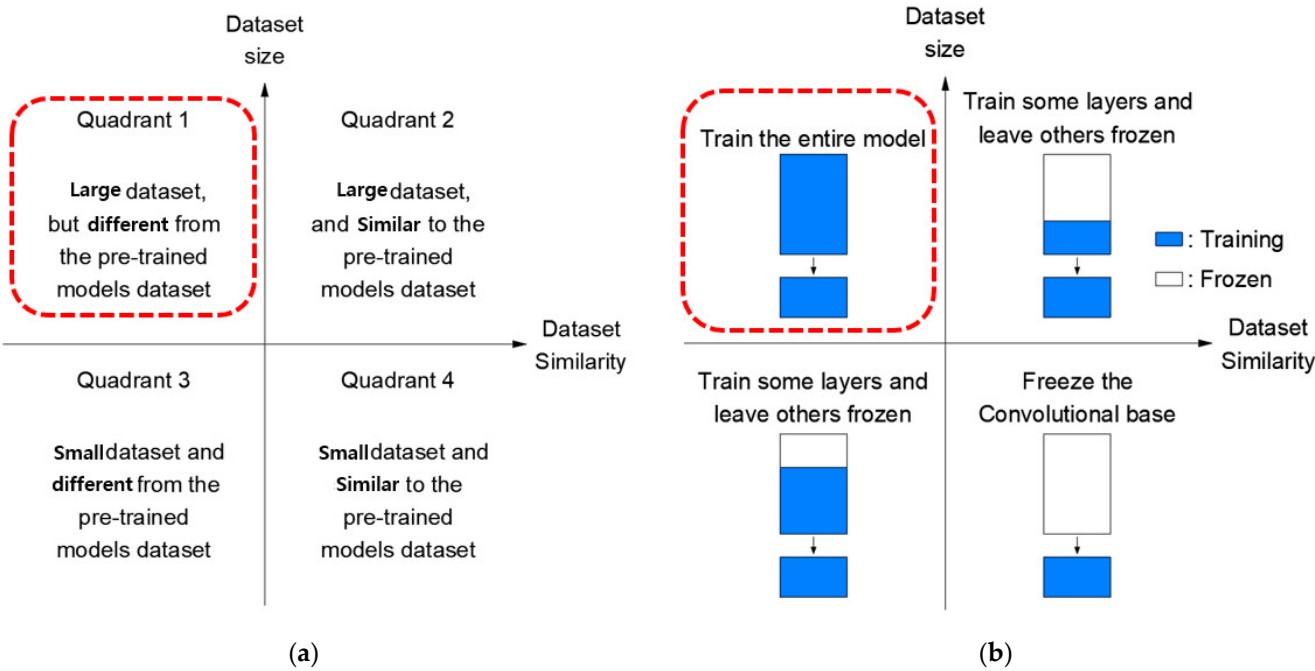

(**a**)            (**b**)

**Figure 2.** Fine-tuning method should be listed as: (**a**) condition of dataset size and similarities; (**b**) options of fine-tuning.

The complexity or number of convolution layers in the current CNN model increases the number of hyperparameters increase, which limits the ability to extract image features and causes overfitting and loss of slope.

To overcome these disadvantages of CNN, Chollet, et al. developed the Xception model by configuring separable convolution layers, as shown in Figure 3. The separable convolution layer method connects the feature maps obtained from the existing convolution layer, obtains a feature map for each channel through depthwise convolution, and then

reduces the number of channels through pointwise convolution. Using this method, the amount of computation of the convolution layer is significantly reduced, and it is possible to reduce the weight of the CNN model [38]. The structure of Xception is shown in Figure 4.

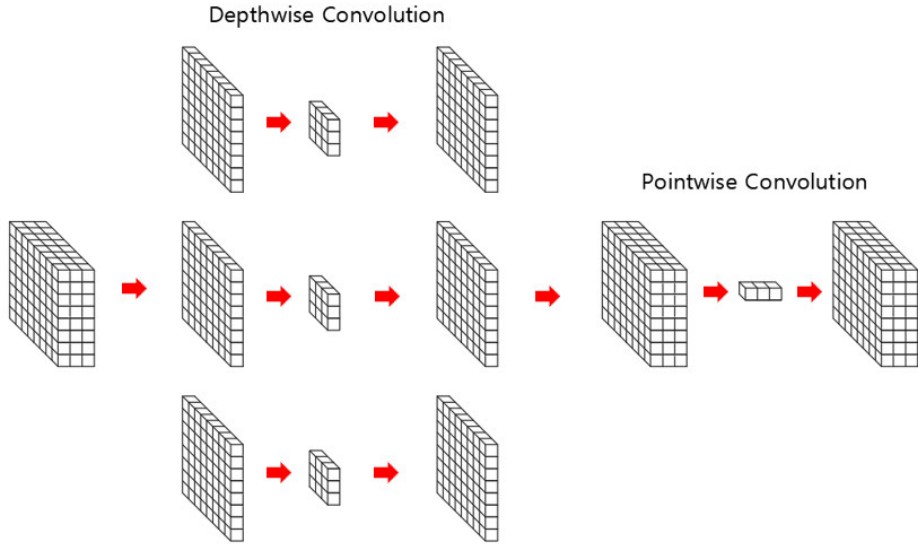

**Figure 3.** Separable layers of CNN.

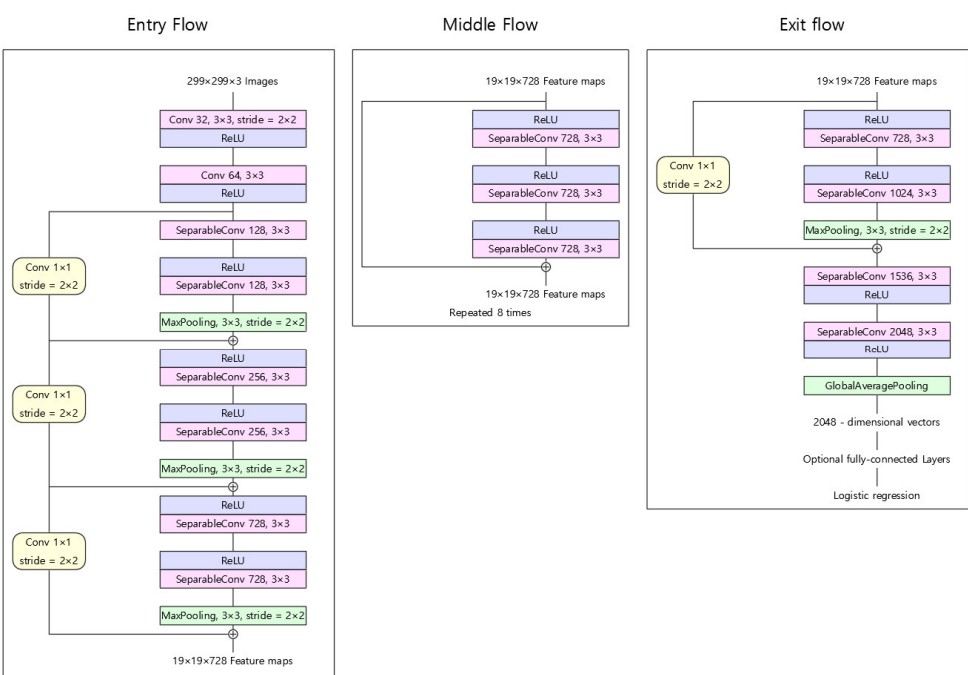

**Figure 4.** Xception model.

## 3. Development of Fault Diagnosis Method Using CNN-Based Transfer Learning

### 3.1. Design of an Artificial Neural Network System

In this study, sound data were directly collected at the workplace and then converted into a spectrogram using the STFT technique. The Xception transfer learning model was then applied to the converted spectrogram to complete the rotor fault diagnosis algorithm. Finally, the completed fault diagnosis algorithm was embedded in the embedded system to perform a validation experiment on the real-time fault diagnosis performance of the rotating body. Figure 5 shows the development procedure for a fault diagnosis system using the CNN-based transfer learning model conducted in this study.

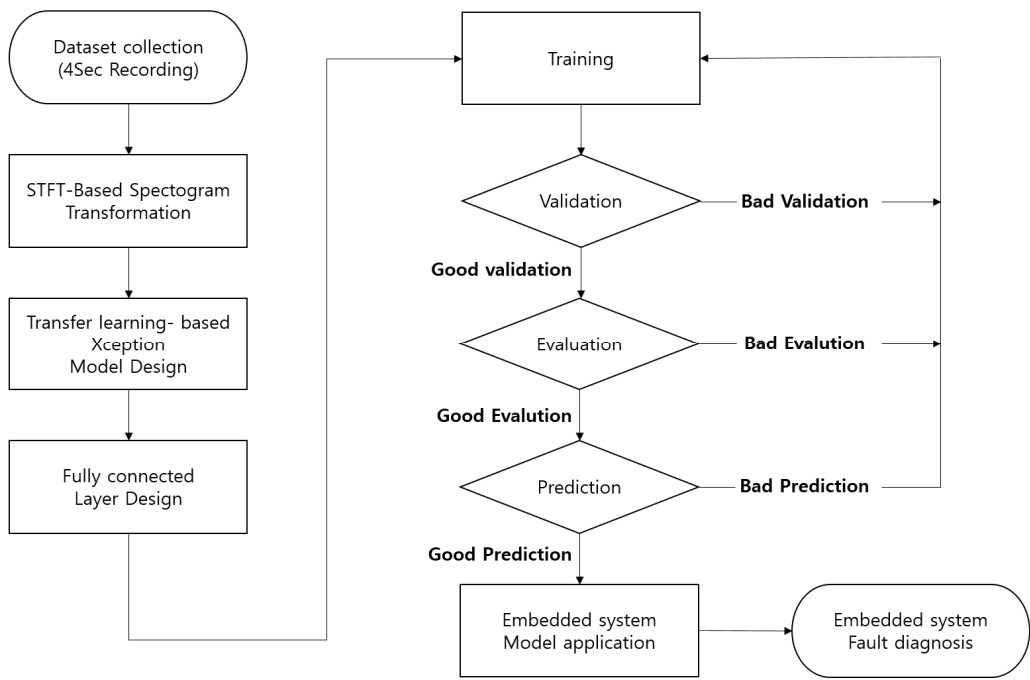

**Figure 5.** Development procedure of a fault diagnosis system using the CNN-based transfer learning model.

### 3.2. Data Collection

We collected various normal and fault sounds of conveyor belt rollers by moving 100 m at a time after recording for 30 min at 4 s intervals outside the quarry and inside the tunnel of a company that produces 7.5 million tons of cement annually in Yeongwol, Gangwon-do. A significant amount of data were collected in the environment where the conveyor belt actually operates to derive the results of artificial intelligence with the same results in the real environment. Figure 6. shows the conveyor belt rollers on the work site where the data were extracted. We obtained 17,000 normal sounds and 17,000 malfunction sounds from conveyor belt rollers. Figure 7. shows the data recorded in 4 s intervals for silent sounds, normal sounds, and trouble sounds.

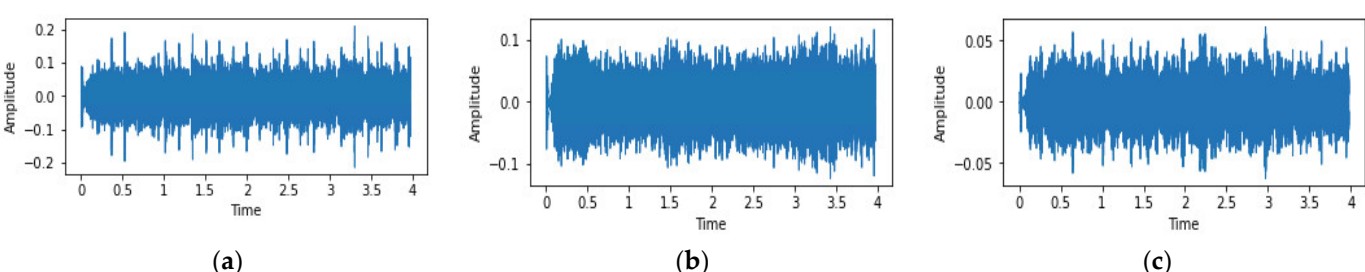

**Figure 6.** Sound signal data should be listed as: (**a**) fault; (**b**) normal; (**c**) silence.

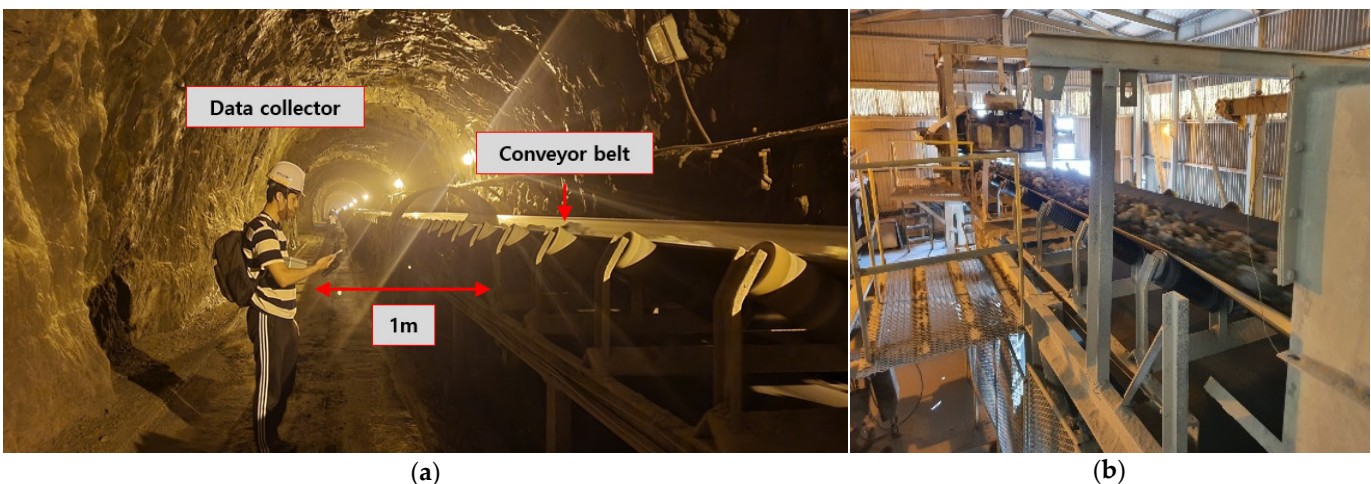

(**a**)                                                                                                  (**b**)

**Figure 7.** Conveyor belt environment, they should be listed as: (**a**) Inside the quarry tunnel; (**b**) Outside the quarry tunnel.

### 3.3. Spectrogram Images Generation

The collected wav files were converted into spectrogram images using the Librosa package in the Python environment, yielding approximately 34,000 spectrogram images, each measuring 128 × 128. Figure 8 shows the conversion of the wave file image into a spectrogram image using the STFT technique. Various patterns of sound information can be confirmed into a spectrogram and visualized using this method for converting sound data generated in the field. In general, a certain pattern frequently appears in the spectrogram when a failure occurs. The length of the sliding window of the spectrogram is 192 GB, and since information for 4 s is mapped, the overlapping size of the spectrogram is set to 512, which is about 1/4 the size, and a hamming window that is easy to analyze continuous sound data used. A total of more than 34,000 spectrogram images were created, and data augmentation was not performed because this was sufficient to perform training, validation, and testing through the CNN model.

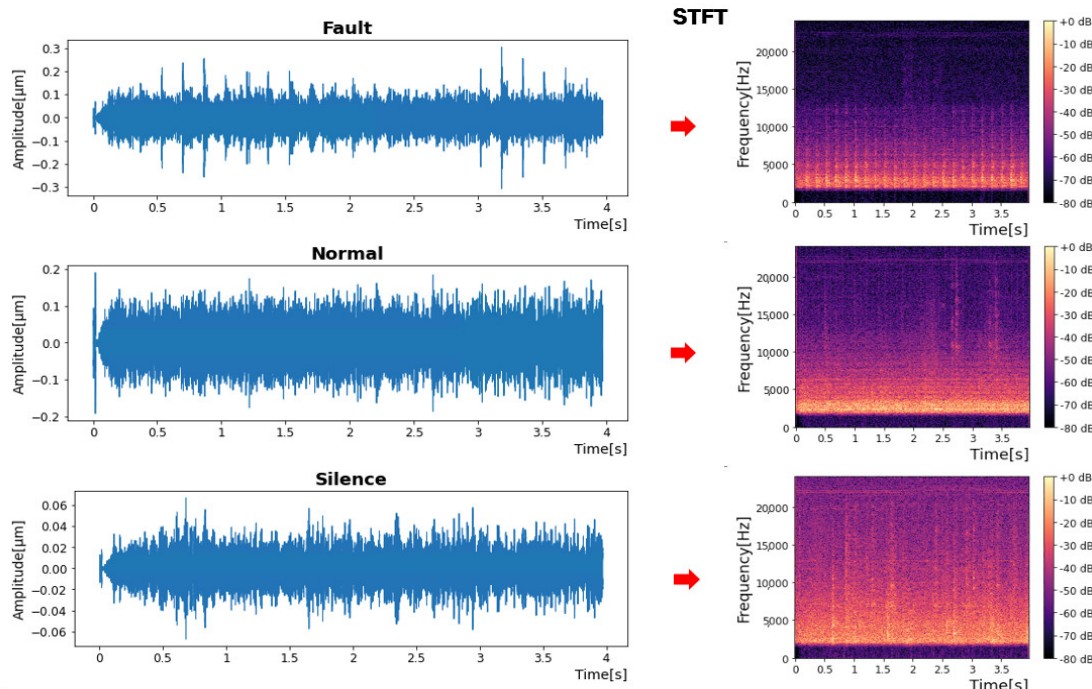

**Figure 8.** Spectrogram image generation.

### 3.4. Xception Model Structure Design

Figure 9 shows the configuration of the CNN-based transfer learning model. The 34,000 datasets applied to the model were divided into groups: training (75%), validation (15%), and testing (10%). The total number of learning epochs was set to 1000, and the learning was automatically terminated early when the loss increased using a callback function. We used model checkpoint, Adam optimization, and binary cross-entropy to implement optimal model learning.

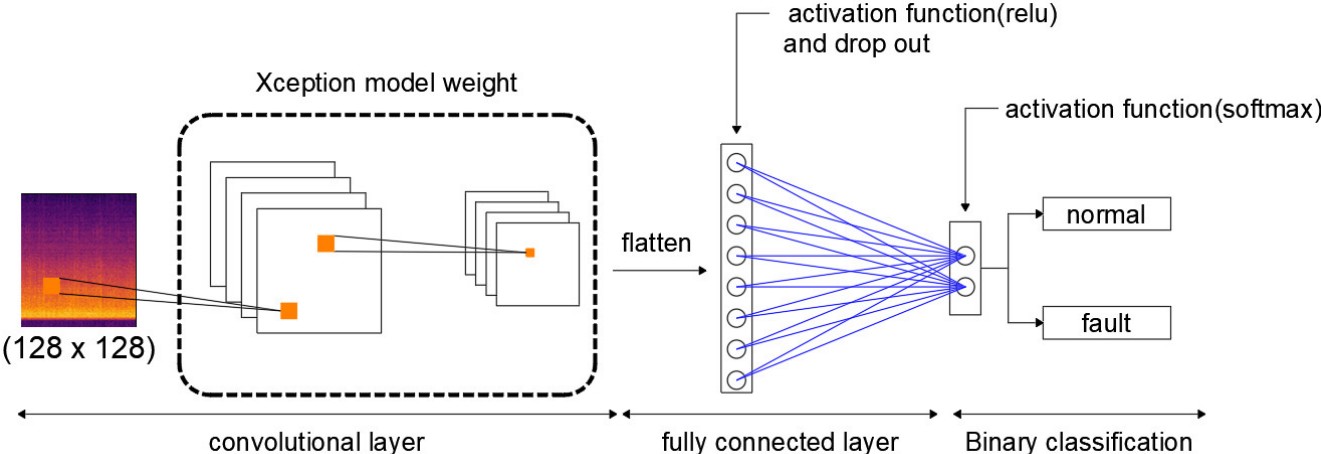

**Figure 9.** Binary classifier based on Xception model.

Table 2 shows the details of the training options of the transfer learning model. CNN training, validation, and evaluation were conducted in a high-capacity computing environment with an NVIDIA GeForce RTX 3090 GPU and 192 GB RAM in the computer environment for developing the CNN-based rotating body fault diagnosis model. The software library performed data preprocessing through Numpy, Pandas, and Matplotlib in Python 3.9.7 environment, and AI training and validation were performed in Tensorflow version 2.5. Conveyor belt sound data processing and spectrogram generation were performed in Pyaudio 0.2.11, Librosa 0.9.1, and OpenCV 4.5.5 versions.

**Table 2.** Train parameters.

| Name of Component | | Content and Value |
|---|---|---|
| Optimizer | | Adam |
| Mini-batch size | | 32 |
| Epoch | | 1000 |
| Loss | | Binary cross entropy |
| Callback | Patience (validation loss) | 10 |
| | Model checkpoint | Best validation accuracy |

## 4. Experimental Results

### 4.1. Configuration of the Embedded System

In this study, a set of experiments to verify the proposed algorithm for diagnosing roller failures was configured as shown in Figure 10. The sound detection module, a unit diagnosis system, is a Raspberry Pi4-based embedded system type controller equipped with a CNN-based Xception model. Since the arrangement of the rollers at the work site is consistently located for a long distance, each embedded model was placed at 3 m intervals corresponding to the maximum separation distance in the laboratory, and a speaker generating the sound of the rollers was placed 1 m in front of the embedded module. Then, using a speaker, the sound data of the normal and faulty state of the roller secured at the actual work site was output, and a diagnostic experiment was performed for 1 h once every 4 s. Additionally, the embedded system developed in this study diagnoses the failure of

the conveyor belt roller sound and transmits the diagnosis result to PCs or PLC equipment through Wi-Fi. Each of the conveyor belt roller sound detection modules distributed around the conveyor belt rotating body collects ambient sound data and performs preprocessing on the collected data. For the experiment, the PWB-05 battery was used as the power source, BM-350U was used as the microphone, and BZ-SP600X was used as the speaker. Then, the sound data that have gone through the preprocessing process are analyzed for the failure of the rotating body by applying an artificial intelligence model. The failure determination result of each module obtained through this process is transmitted to the upper controller, and instead of the entire existing sound data, only the failure determination result is transmitted, thereby reducing the load concentrated on the PLC or PCs. Figure 11 hows the internal equipment of the embedded system used in this experiment.

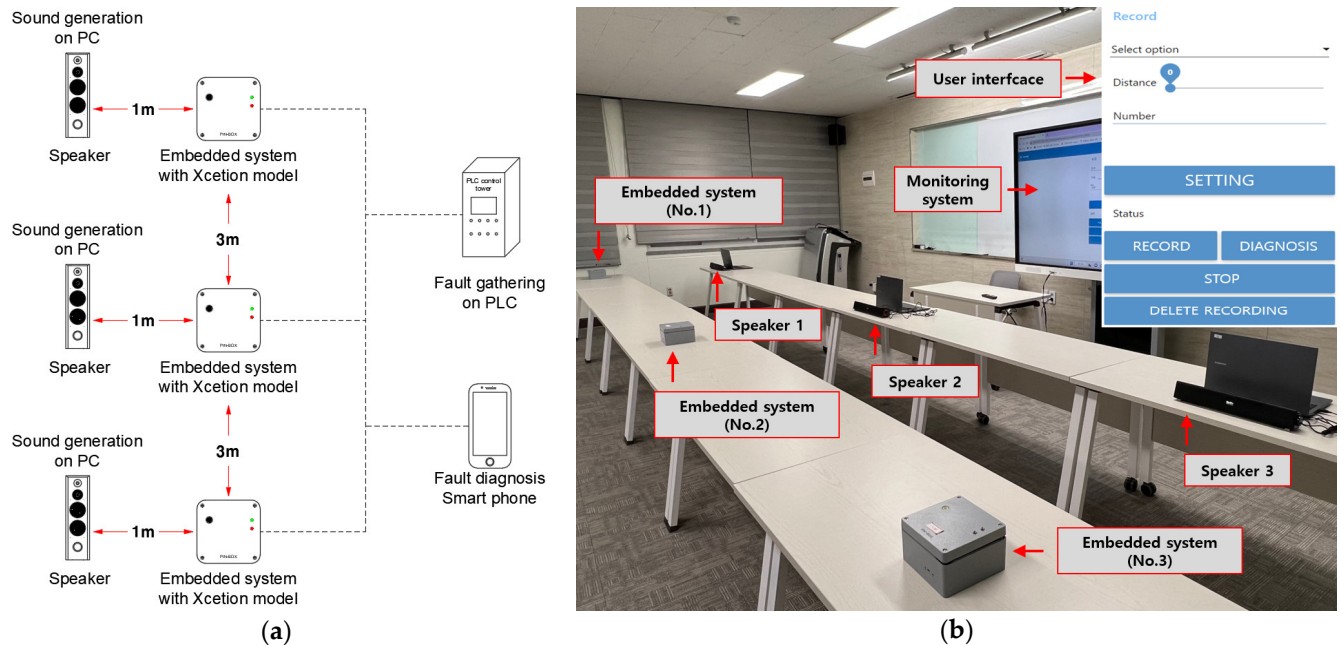

**Figure 10.** Embedded systems experimental environment as: (**a**) embedded system schematic; (**b**) embedded systems installed inside the laboratory.

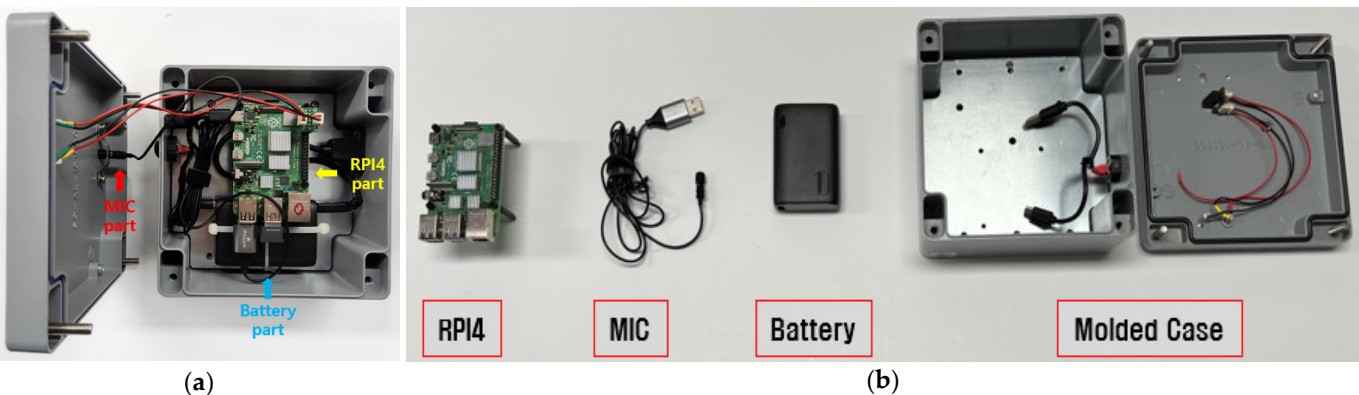

**Figure 11.** Internal structure of embedded system devices as: (**a**) embedded system internal structure; (**b**) embedded system internal equipment.

Figure 12 shows a schematic diagram of an embedded software system. The conventional diagnosis of abnormalities in the rolling element required a worker to visit the site and analyze the sound and condition of the roller. Therefore, the implementation of wireless communication between a system that diagnoses the presence of abnormalities

in the field in real time and a remote supervisor's monitoring system is a very important issue. In this research, we introduced the RPI4 embedded system to build these environments. Embedded systems now include servers such as Dnsmasq, Hostapd, etc., for remote connections, enabling wireless local networks. The way the whole system works is designed with a GUI based on Node-Red. Rotating body sound data measured in real time was used to determine the normality and failure of the conveyor belt through an artificial intelligence model installed in RPI4, and the determination results were stored in a USB memory. At the same time, the discrimination result can be checked in real time on a remote PC through MQTT. The user interface outputs the result of determining whether it is normal or malfunctioning, the distance between the roller and the recorder, and the number of recordings.

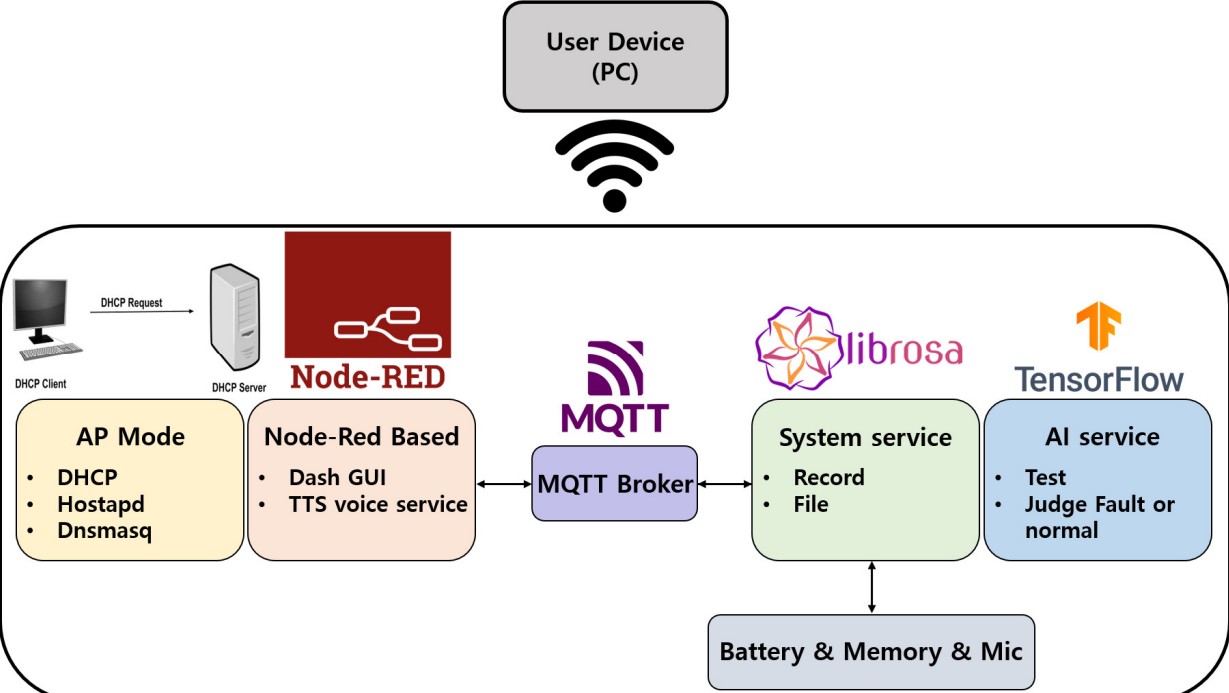

**Figure 12.** Embedded software system schematic.

*4.2. Results of Xception Model Learning*

Figure 13 shows the validation accuracy and validation loss according to the number of epochs during the learning process. The weights of models with the highest validation accuracy during the learning process were stored at 396 epochs, and learning was stopped early at 406 epochs out of a 1000 epoch set. The developed fault diagnosis transfer learning model showed a high validation accuracy of 99.567% and a low validation loss of 0.014.

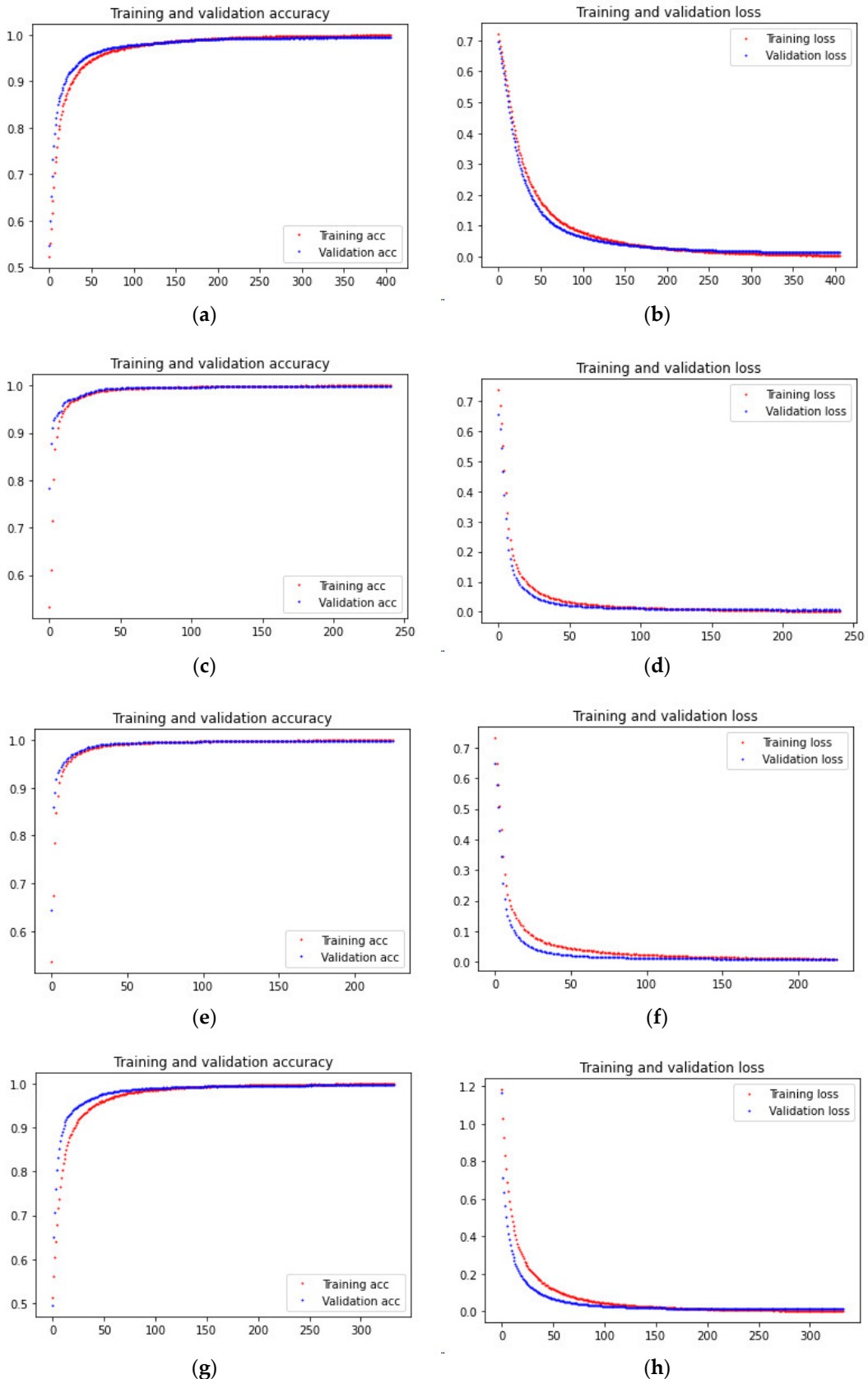

**Figure 13.** Comparison of training and validation results in various transfer learning models should be listed as: (**a**) Xception accuracy; (**b**) Xception loss; (**c**) VGG16 accuracy; (**d**) VGG16 loss; (**e**) VGG19 accuracy; (**f**) VGG19 loss; (**g**) ResNet50 accuracy; (**h**) ResNet50 loss.

Figure 14 shows the confusion matrix for model prediction, and the classification report in Table 2 shows the precision, recall, and harmonic average (F1-score). Equations (6) to (8) are used to express precision, recall, accuracy, and harmonic mean, respectively.

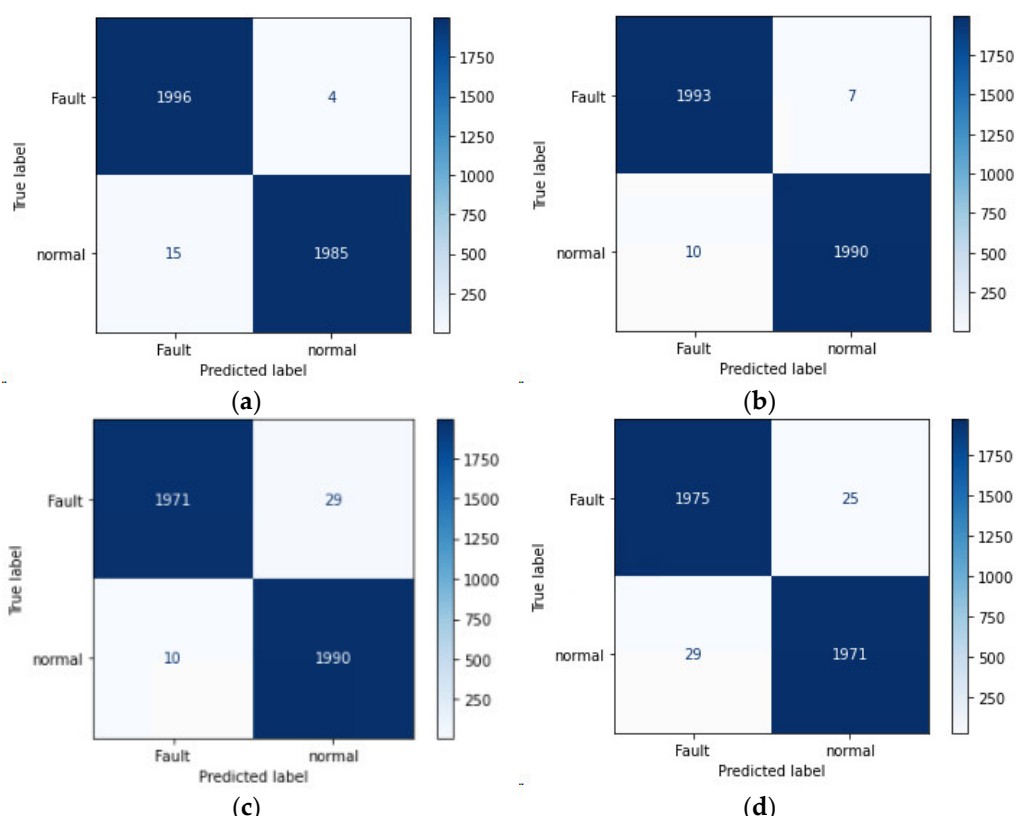

**Figure 14.** Prediction of Confusion Matrix, they should be listed as: (**a**) Xception Test dataset; (**b**) VGG16 Test dataset; (**c**) VGG19 Test dataset; (**d**) ResNet50 Test dataset.

In the model prediction results in Table 2, the precision, recall, accuracy, and harmonic mean of the validation and test datasets were all at least 98%.

As shown in Figure 13, overfitting did not occur during learning in this experiment. Additionally, the validation error gradually decreases, and the validation accuracy tends to increase.

The reason why overfitting did not occur was that sufficient datasets were used in this experiment, and the spectrogram image data were set to 0 and 1 classes through one-hot encoding. In addition, unbiased dataset distribution was confirmed through rigorous data preprocessing, and training and validation were performed. In addition, dropout was applied to reduce the complexity of the hyperparameter of the fully connected layer, and early stopping was applied to automatically end training when the validation loss reduction did not proceed within 10 epochs of the learning curve of the model.

$$Precision = \frac{TP}{TP + FP} \tag{6}$$

$$Recall = \frac{TP}{TP + FN} \tag{7}$$

$$F1\ Score = \frac{Precision \times Recall}{Precision + Recall} \tag{8}$$

In the validation dataset, 2992 of 3000 images from the fault dataset were determined as faults, and 2997 of 3000 images from the normal dataset were determined as normal. Regarding the test dataset, 1996 of 2000 images from the fault dataset were determined

as faults, and 1985 of 2000 images from the normal dataset were determined as normal. Although the accuracy of the test dataset is higher regarding the normal and fault, as 97.8% and 9.99%, respectively, it is less accurate than the validation dataset in terms of prediction results. The accuracy of the test dataset was lower than that of the validation dataset in the terms of prediction. However, the accuracy of the test dataset was with 97.8% and 9.99% accuracy for normal and malfunction, respectively.

As shown in Table 3, all four CNN models training results, precision, recall, accuracy, and F1-score, produced high results. In order to apply to RPI4, a small computer for embedded systems, the Xception model with the smallest model capacity and hyperparameters was applied to the embedded system and experiments were conducted.

**Table 3.** Test dataset classification results.

| Name | | Precision | Accuracy | Recall | F1-Score | Total Data No. |
|---|---|---|---|---|---|---|
| Xception | normal | 0.993 | 0.995 | 0.998 | 0.995 | 2000 |
| | fault | 0.998 | | 0.993 | 0.995 | 2000 |
| VGG16 | normal | 0.995 | 0.996 | 0.997 | 0.996 | 2000 |
| | fault | 0.997 | | 0.995 | 0.996 | 2000 |
| VGG19 | normal | 0.995 | 0.990 | 0.986 | 0.990 | 2000 |
| | fault | 0.986 | | 0.995 | 0.990 | 2000 |
| ResNet50 | normal | 0.986 | 0.986 | 0.988 | 0.987 | 2000 |
| | fault | 0.988 | | 0.986 | 0.987 | 2000 |

*4.3. Result Applied to the Embedded System*

Table 4 shows the results of the failure diagnosis experiment using the developed model and the embedded system. Although the developed model showed low accuracy compared to the test dataset, it showed high accuracy of 94.89–99.44% when used on an embedded system similar to the actual working environment.

**Table 4.** Accuracies according to the sound conditions.

| Name | Accuracy [%] | Total Data No. | True | False |
|---|---|---|---|---|
| Normal | 94.89 | 900 | 854 | 46 |
| Fault | 99.44 | 900 | 895 | 5 |

Figure 15 shows the result of fault determination for 900 fault sound data for 1 h in 6 min intervals using the embedded system. Only 5 of 900 pieces of data showed judgment errors.

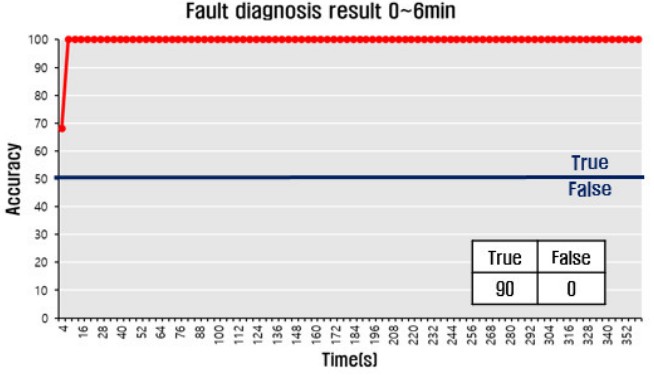
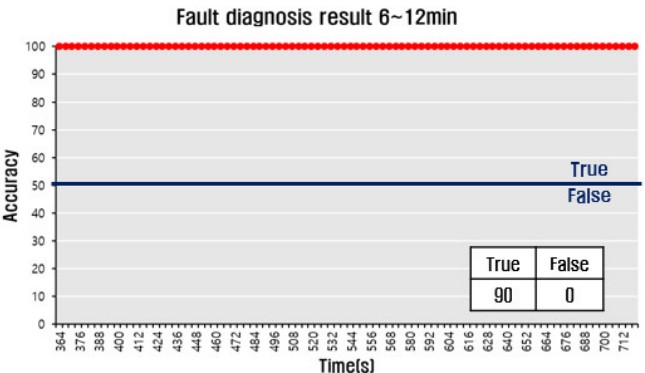

**Figure 15.** *Cont.*

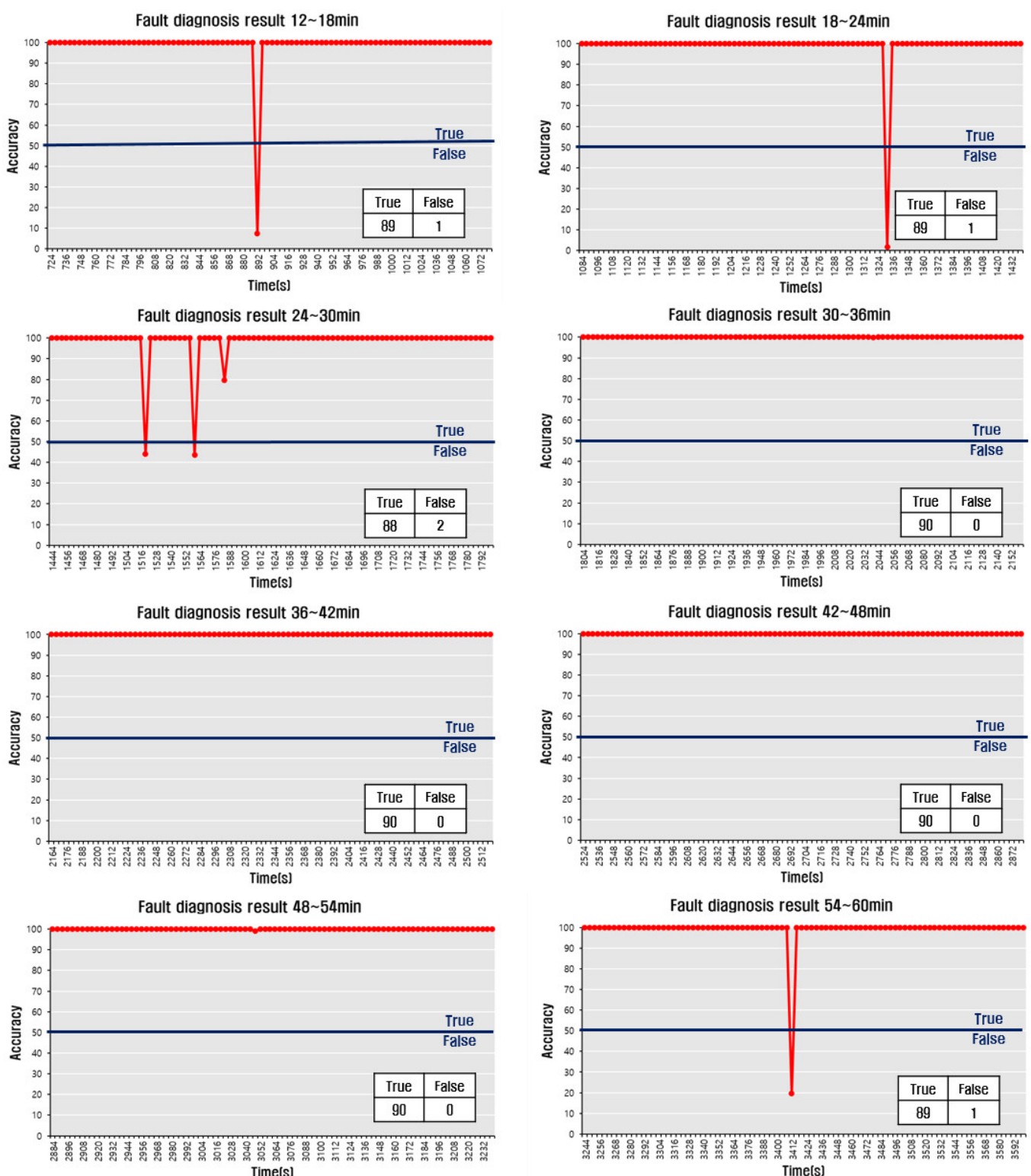

**Figure 15.** Test result with fault sound for 0~60 min.

Figure 16 shows a relatively low accuracy of 94.89% by discriminating 854 of the 900 normal sounds of the conveyor belt as normal. However, it was confirmed that the figure of 94.89% was also high in accuracy and showed good results for normal sound data. However, it is estimated that a fine failure signal is generated in normal sound data because of the lower accuracy than the failure sound data. To address this, it is necessary to further strengthen data preprocessing or refine classification to improve the accuracy

of diagnostic systems in addition to the faults and normals due to conventional binary classification. Laboratory-level experiments on conveyor belt defect sound data have proven high accuracy for normal and fault diagnosis, and high utilization at the actual conveyor work site can also be expected in the future.

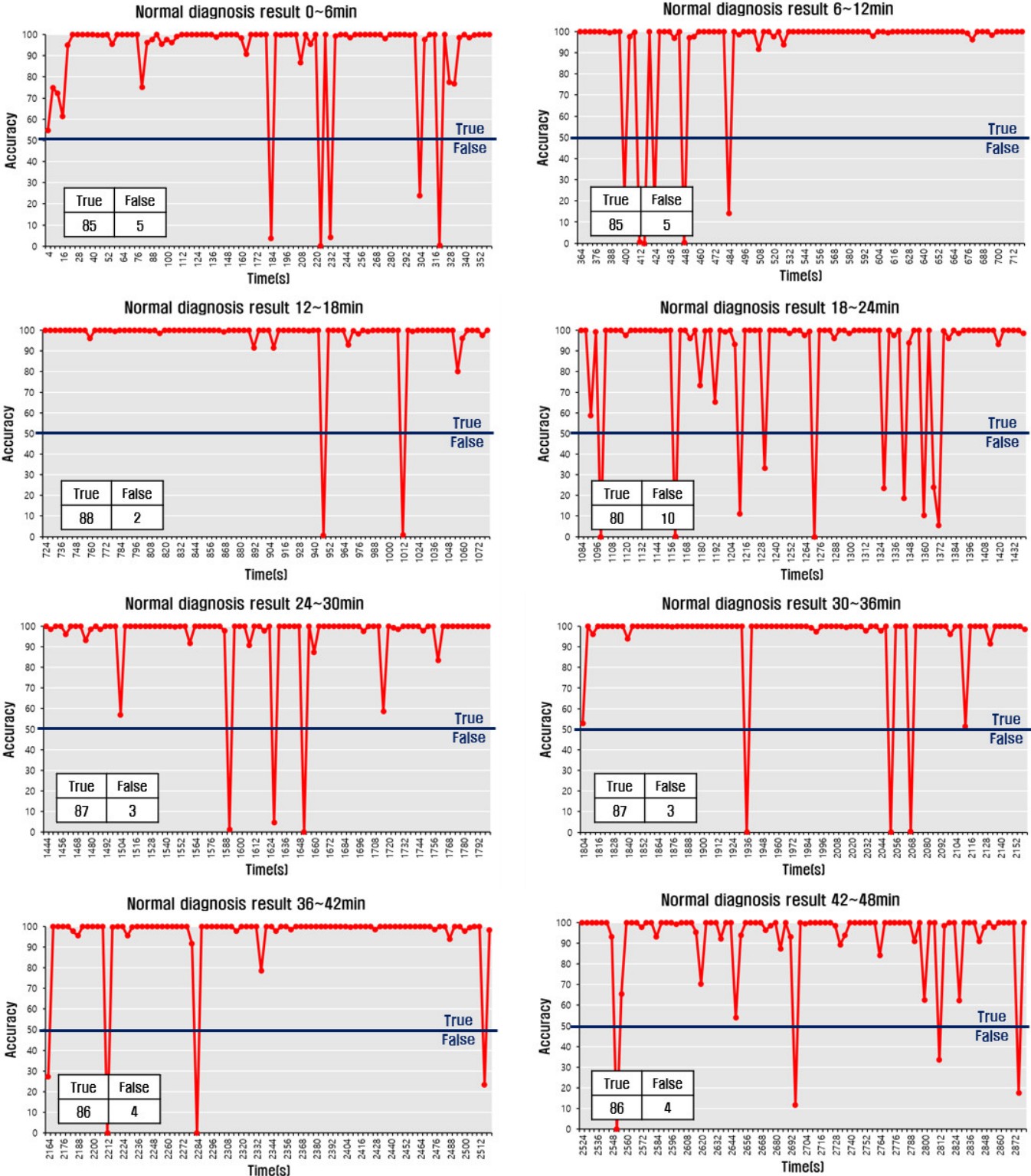

**Figure 16.** *Cont.*

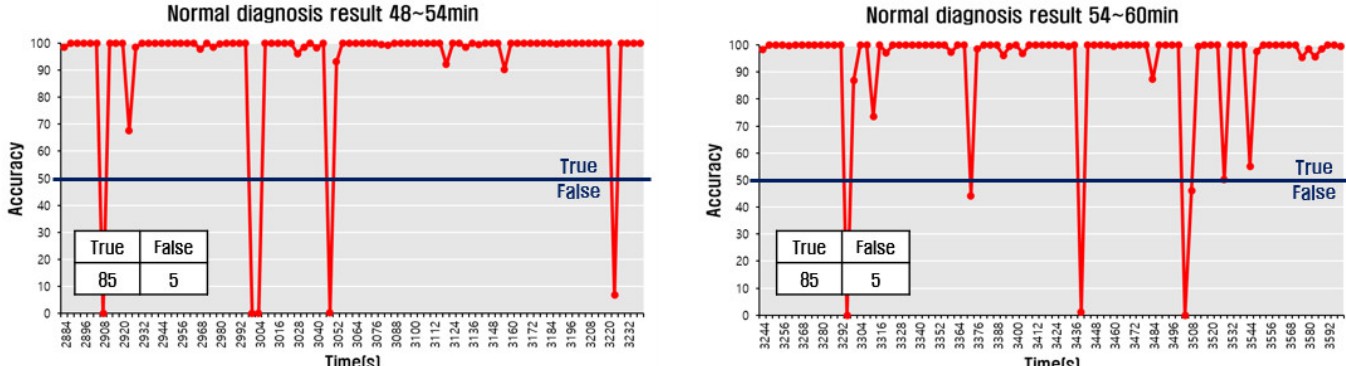

**Figure 16.** Test result with normal sound for 0~60 min.

## 5. Conclusions and Future Research Plans

This study proposed a fault diagnosis method using CNN that converts various sounds generated from rotors such as conveyor belt rollers in industrial sites into spectrogram through STFT techniques and applies complex patterns of visualized images. The sound data generated at the work site was directly collected to create a dataset necessary for deep learning, and various preprocessing tasks, including spectrograms, were performed on the collected data to be suitable for deep learning. Additionally, we used the Xception transfer learning model to improve the classification accuracy using little data. Because the transfer learning model is generalized, a fully connected layer and a classifier were independently designed and applied to the transfer learning model to create a structure appropriate for the rotor fault site. The developed model was applied to a Raspberry Pi-based embedded system to perform a fault diagnosis experiment, and the analysis was conducted using various matrices for the experimental results. Using the results of the experiment, it was possible to confirm that more than 99% of diagnoses for various faults and normal sounds and typical noises were accurate. In conclusion, this study performed model learning based on work site data and developed a rotor fault diagnosis system using the STFT technique and transfer learning model. It was confirmed that the developed artificial neural model can determine various normal and fault sounds at the work site with a high level of accuracy using an embedded system.

In addition, conventional CNN models cannot demonstrate high accuracy in diagnosing new failure modes that occur such as conveyor belt production lines with a lot of field noise and various nonlinear factors. So, we proposed a lightweight CNN model and real-time testing of the CNN model. Based on the findings of this study, it is expected that the current manual defect diagnosis method, in which a person directly inspects a conveyor belt for defects, will reduce the number of human casualties, such as worker pinching accidents.

Additionally, this proposed method has the advantage of obtaining field information remotely by processing field discrimination data in real time in an artificial intelligence-based discrimination system installed on the site, away from the conventional method of manually visiting the site and manually determining the failure. Simplification of information processing is expected as it directly judges the raw failure information data in the field without having to process it again. As a result of this study, it tends to detect well when the fault sound occurs clearly in the field, but it is classified as normal in the early phase of the fault where the fault sound is weak or occurs very intermittently. In this case, in order to detect the early stage of failure, it is necessary to study a classification technique that learns by further subdividing the stage of failure into early, middle, and late stages of failure. In addition, we plan to conduct research on the development of a failure diagnosis system applicable to the field by analyzing the real-time diagnosis results for classes such as normal and failure for various sound data generated from the rotating body of the conveyor belt in the actual work site. In addition to diagnosing the failure of the rotating body, we plan to conduct research on the development of a big data analysis

system and artificial intelligence-based prediction model that can predict the replacement cycle due to deterioration, damage, and looseness of the rotating body at an early stage.

In future research, we intend to improve the binary classification method of failure and normal and to develop an artificial intelligence algorithm that can diagnose more detailed failures by categorizing different failure types into early, middle, and late stages.

**Author Contributions:** Conceptualization, H.J. and S.C.; methodology, H.J.; software, S.C.; validation, H.J., S.C. and B.L.; formal analysis, H.J.; investigation, H.J. and S.C.; data curation, H.J.; writing—original draft preparation, H.J.; writing—review and editing, H.J. and S.C.; supervision, S.C. and B.L.; project administration, H.J. and B.L.; funding acquisition, H.J. All authors have read and agreed to the published version of the manuscript.

**Funding:** This paper was supported by the Semyung University Research Grant of 2020.

**Informed Consent Statement:** Informed consent was obtained from all subjects involved in the study.

**Data Availability Statement:** Not applicable.

**Conflicts of Interest:** The authors declare no conflict of interest.

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
