# Peer review of "Rotor Fault Diagnosis Method Using CNN-Based Transfer Learning with 2D Sound Spectrogram Analysis"

_electronics, doi:10.3390/electronics12030480_

Round 1

Reviewer 1 Report

In general, the text is very well structured and has clearly defined topics. The abstract is a very good guide for what follows. More or less all fundamental theory details that are needed are discussed and concluding remarks are sufficient. The authors have made a concise overview of the topic and a brief reference to existing literature. They have indicated the main task of the paper among its motivation. Finally, they have pointed out the key message and the potential benefits of their work.

Some comments for improvement:

1. As a general drawback I could say that there is no reference to similar approaches (e.g. [1], [2]) where the same techniques have been used for the oil and gas and shipping industries. Besides in works like [3] and [4] authors try to approach similar issues using different supervised and unsupervised methods.

[1] Spandonidis, C., Theodoropoulos, P., Giannopoulos, F., Galiatsatos, N., & Petsa, A. (2022). Evaluation of deep learning approaches for oil & gas pipeline leak detection using wireless sensor networks. Engineering Applications of Artificial Intelligence, 113, 104890.

[2] Theodoropoulos, P., Spandonidis, C. C., & Fassois, S. (2022). Use of Convolutional Neural Networks for vessel performance optimization and safety enhancement. Ocean Engineering, 248, 110771.

[3] Hou, Y., He, R., Dong, J., Yang, Y., & Ma, W. (2022). IoT Anomaly Detection Based on Autoencoder and Bayesian Gaussian Mixture Model. Electronics, 11(20), 3287.

[4] Ahn, H., & Yeo, I. (2021). Deep-Learning-Based Approach to Anomaly Detection Techniques for Large Acoustic Data in Machine Operation. Sensors, 21(16), 5446.

2. Justification regarding the selection of the algorithms mentioned in 2.3 should be provided. Did the authors test different models? LSTM RNN or 1D CNN could be a good fit for the problem under discussion.

3. The introductory paragraph in section 4 should be refined such that the reasoning behind the lab scale test is better presented.

4. Based on the previous comment the research questions and the added value of the work could be better clarified in section 1.

5. Did authors test their results against other methods? Comparisons with similar or different methods could provide valuable insights regarding the proposed method’s efficiency.

6. Concluding remarks should also include the impact of the work and its novelty.

Author Response

We are sincerely grateful for your thorough consideration and scrutiny of our manuscript,

“Rotor fault diagnosis method using CNN-based transfer learning”, Manuscript ID: electronics-2099390

Reviewer 2 Report

The manuscript entitled “Rotor fault diagnosis method using CNN‐based transfer learning” proposes a method for detecting rotor faults based on spectrograms of faulty rotor sound signals used as input to the CNN‐based classifier utilizing the transfer learning principle. The proposed method is experimentally validated on real-world data, showing promising results.

Here are some comments I would like the authors to address in the revised manuscript:

1.      Please include the word “spectrogram” in the title of the manuscript, as it is an essential part of the proposed detection method. This will provide a more precise title.

2.      The literature review is very limited, mentioning only a few references, mostly without discussing their conclusions and contributions. The authors should extend the literature review by addressing several more recent references in the field of CNN-based rotor fault detection, thus placing the manuscript within this research field.

3.      Moreover, the application of deep CNNs with various 2D time-frequency signal representations, including STFT spectrograms, has recently become a hot research topic. Therefore, I would like to suggest the authors supplement the introductory part with some of the recent studies on this topic to briefly illustrate the state-of-the-art performances of the CNNs and time-frequency representations in many different applications today, in addition to those in rotor fault detection, and provide an interested reader with examples. Please consider briefly mentioning the following papers for illustration purposes: 10.3390/rs12101685, 10.1109/ACCESS.2021.3139850, 10.3390/app11083603.

4.      All equations in the manuscripts are blurry. This needs to be corrected.

5.      The authors should provide details about the spectrogram generation procedure. Which type of sliding window and which size of the window were utilized?

6.      The first row in Table 2 is not very informative. Please correct.

7.      In my opinion, there is no need to put the software and hardware specifications in table form (Tables 3 and 4). This can be given in a few lines of the manuscript’s main text.

8.      Why was a balanced dataset (50% normal and 50% faulty sounds) considered in this binary classification problem if faulty rotors are expected to occur less frequently?

9.      Figures 10 and 11 do not have captions. Please correct.

10.  Did the authors consider using any techniques for data augmentation?

11.  The reported classification metrics are very high. The authors should provide additional elaboration that no overfitting occurred during the training.

12.  Why are there no accuracy values provided in Table 5?

13.  In my opinion, there is no need to compare the results obtained on the test dataset with those obtained during model validation. The results on the test dataset are what matters.

14.  In the Conclusion section, please address the limitations of the proposed method.

15.  In the Conclusion section, please elaborate more on directions for future research.

Author Response

(The authors gave the same response as above.)

Round 2

Reviewer 2 Report

The authors have addressed most of my comments.

However, the response to Comments 2 and 3 is not satisfactory. The literature review has not been adequately extended. Although the authors have added some references, these were related to the detection of faults and malfunctions in other fields (e.g., IoT, ships, pipelines), and not to rotor fault detection.

Moreover, the authors have not extended the literature review regarding the general application of CNNs with 2D time-frequency representations to provide appropriate theoretical background and illustrate with examples. Only one of the suggested references has been added, and others have been disregarded.

Finally, much more attention needs to be paid to the literature review, according to the above comments, in the revised version of the manuscript.

Author Response

Dear reviewer 2

We are sincerely grateful for your thorough consideration and scrutiny of our manuscript.

Through the accurate comments made by the reviewers, we better understand the critical issues in this paper.      We have revised the manuscript according to the Editor’s and reviewer’s suggestions. We hope that our revised manuscript will be considered and accepted for publication in the Journal of Electronics. We acknowledge that the scientific and clinical quality of our manuscript was improved by the scrutinizing efforts of the reviewers and editors.

Below, we address the reviewer’s comments and list the changes that we made to the manuscript according to their reports. The original reviewer’s comments are provided in black, whereas ours are given in red.

All revisions to the manuscript were marked up using the "Tracking Change" function. Point-by-point responses to the reviewers’ comments are provided below.

We attached the revision file.

Once again, we must say that we are very much obliged to you.

Sincerely,

Haiyoung, Jung

Round 3

Reviewer 2 Report

The authors have addressed my comments.

Author Response

Dear editorial staffs in Journal of Electronics

We are sincerely grateful for your thorough consideration and scrutiny of our manuscript,

“Rotor fault diagnosis method using CNN-based transfer learning with 2-D sound spectrogram analysis”, Manuscript ID: electronics-2099390.

Through the accurate comments made by the reviewers, we better understand the critical issues in this paper.      We have revised the manuscript according to the Editor’s and reviewer’s suggestions. We hope that our revised manuscript will be considered and accepted for publication in the Journal of Electronics. We acknowledge that the scientific and clinical quality of our manuscript was improved by the scrutinizing efforts of the reviewers and editors.

Below, we address the reviewer’s comments.

All revisions to the manuscript were marked up using the "Tracking Change" function.

In particular, the missing reference papers in the last revision file were correctly written.

Also, the parts that do not match the template have been corrected.

We would be happy to make any further changes that may be required

Once again, we must say that we are very much obliged to you.

We look forward to hearing from you.

Thank you and Best Regards,

Haiyoung, Jung
